# Bulk-local-density-of-state correspondence in topological insulators

Biye Xie[1,2,7], Renwen Huang[3,4,7], Shiyin Jia[3,4,7], Zemeng Lin[1,7], Junzheng Hu[3,4], Yao Jiang[3,4], Shaojie Ma [1], Peng Zhan [3,4] ✉, Minghui Lu [3,5] ✉, Zhenlin Wang [3,4], Yanfeng Chen [3,5] & Shuang Zhang [1,6] ✉

In the quest to connect bulk topological quantum numbers to measurable parameters in real materials, current established approaches often necessitate specific conditions, limiting their applicability. Here we propose and demonstrate an approach to link the non-trivial hierarchical bulk topology to the multidimensional partition of local density of states (LDOS), denoted as the bulk-LDOS correspondence. In finite-size topologically nontrivial photonic crystals, we observe the LDOS partitioned into three distinct regions: a two-dimensional interior bulk area, a one-dimensional edge region, and zero-dimensional corner sites. Contrarily, topologically trivial cases exhibit uniform LDOS distribution across the entire two-dimensional bulk area. Our findings provide a general framework for distinguishing topological insulators and uncovering novel aspects of topological directional band-gap materials, even in the absence of in-gap states.

Topological materials, which transcend the conventional spontaneous symmetry-breaking paradigm, have significantly advanced our comprehension of condensed matter physics[1–7]. They hold great promise for applications in energy-efficient electronics[8] and quantum computing[9] owing to their robust and distinctive transport properties[10–17]. Theoretical descriptions of topological phases rely on quantized bulk physics known as topological invariants[18,19]. However, experimental measurement of these bulk topological invariants is challenging and typically demands sophisticated quantum state reconstruction[20]. Fortunately, correspondences exist between bulk topological physics and other experimentally accessible observables. For example, in the case of completely gapped topological phases (or topological insulators, TIs), conventional bulk-boundary correspondence (BBC)[21] dictates the presence of in-gap topological boundary states at the interface between two topologically distinct materials[22,23]. The powerful BBC has played a pivotal role in characterizing TIs and has been extended to higher-order topology[24], non-Hermitian

topology[25], and 4D quantum Hall systems[26]. However, recent studies on topological crystalline insulators[27–32] have revealed that topological phases without chiral symmetry (or particle–hole symmetry) can habor boundary states embedded within the bulk spectrum. Consequently, the BBC fails to precisely distinguish these specific topological phases.

To address this challenge, Peterson et al.[33] introduced a measurable topological indicator for identifying non-trivial higher-order topological crystalline phases, eliminating the need for in-gap localized boundary states. Specifically, they quantified the portion of Wannier centers (WCs), also known as spectral charges, by integrating the local density of states (LDOS) within each unit cell in the bulk, edge, and corner regions across the entire band spectrum. This approach defined a fractional corner anomaly (FCA) that links non-trivial higher-order topological insulating phases to observable fractional quantum numbers. More recently, leveraging similar measurements of the fractional quantum number (associated with the count of WCs within a

[1]New Cornerstone Science Laboratory, Department of Physics, The University of Hong Kong, Pokfulam Road, Hong Kong, China. [2]School of Science and Engineering, The Chinese University of Hong Kong, 518172 Shenzhen, China. [3]National Laboratory of Solid State Microstructures, Collaborative Innovation Center of Advanced Microstructures, Nanjing University, 210093 Nanjing, China. [4]School of Physics, Nanjing University, 210093 Nanjing, China. [5]Department of Materials Science and Engineering, Nanjing University, 210093 Nanjing, China. [6]Department of Electrical and Electronic Engineering, University of Hong Kong, Hong Kong, China. [7]These authors contributed equally: Biye Xie, Renwen Huang, Shiyin Jia, Zemeng Lin. ✉e-mail: zhanpeng@nju.edu.cn; luminghui@nju.edu.cn; shuzhang@hku.hk

unit cell), a bulk-disclination correspondence has been established to predict the existence of topological disclination states with various spatial symmetries[32,34]. However, a fundamental limitation persists in this correspondence: WCs can only be defined (and subsequently measured) across an entire band (or bands), and their distribution within unit cells is significantly influenced by lattice spatial symmetries. For a broad class of topological insulators (TIs) characterized by directional bandgaps, sometimes referred to as partial bandgaps[35] and lacking in-gap states or experiencing disorders that disrupt all spatial symmetries, both the BBC and FCA fail to characterize these topological phases. Therefore, a pivotal question arises: What is the general correspondence between non-trivial bulk topology and measurable observables?

In this study, we introduce and validate an approach for the precise identification of distinct topological phases characterized by Wannier centers (WCs). We achieve this by examining the multidimensional and single-dimensional partition of the local density of states (LDOS) for topologically nontrivial and trivial phases, thereby establishing a comprehensive link between non-trivial bulk topology and measurable LDOS. To mitigate finite-size effects, we calculate the averaged LDOS across a narrow energy range and over bulk, edge, and corner regions for both topologically nontrivial and trivial lattices. Remarkably, we find that the magnitudes of the averaged LDOS in this region remain nearly constant across energy levels for trivial phases, whereas they exhibit significant variations for nontrivial phases. We provide an intuitive explanation for this phenomenon, drawing from modern polarization theory[31] and demonstrate its validity through experiments conducted in a two-dimensional (2D) photonic system characterized by higher-order topological insulators (TIs) featuring

directional bandgaps and devoid of in-gap localized boundary states. Furthermore, we establish the robustness of this correspondence in systems affected by random disorders. Our findings furnish a universal criterion for diagnosing topological phases and open avenues for the investigation of topological phases within materials possessing topological directional bandgaps.

## Results

### Multidimensional partition of LDOS reveals topology

To gain an intuitive grasp of this approach, we commence with the concept of a topological insulator (TI), where the topological phases are distinguished by the displacement of Wannier centers (WCs) within unit cells, signifying bulk polarizations and filling anomalies[31,36–38]. In TIs, WCs are situated away from the centers of unit cells. Specifically, WCs positioned at the sides and corners of unit cells correspond to first-order and second-order TIs, respectively, as depicted in Fig. 1a. Conversely, if WCs are found at the centers of unit cells, the system is a trivial atomic topological insulator, often referred to as an ordinary insulator (OI) (see Fig. 1b). Since WCs represent charge centers of wavefunctions[39], the number of WCs in specific regions governs the magnitude of the local density of states (LDOS) in those areas. Moreover, in a finite-size lattice, WCs located at bulk, edge, and corner areas (highlighted in colored regions in Fig. 1) dictate the portion of wavefunctions that combine to form bulk, edge, and corner states, respectively. Consequently, for topologically nontrivial lattice with WCs localized at the corner of unit cells as shown in Fig. 1a, the multidimensional hybridization of single lattice site's orbitals at bulk (green area in Fig. 1a), edge (blue area in Fig. 1a), and corner areas (red area in Fig. 1a) results in the multidimensional partition of LDOS in the

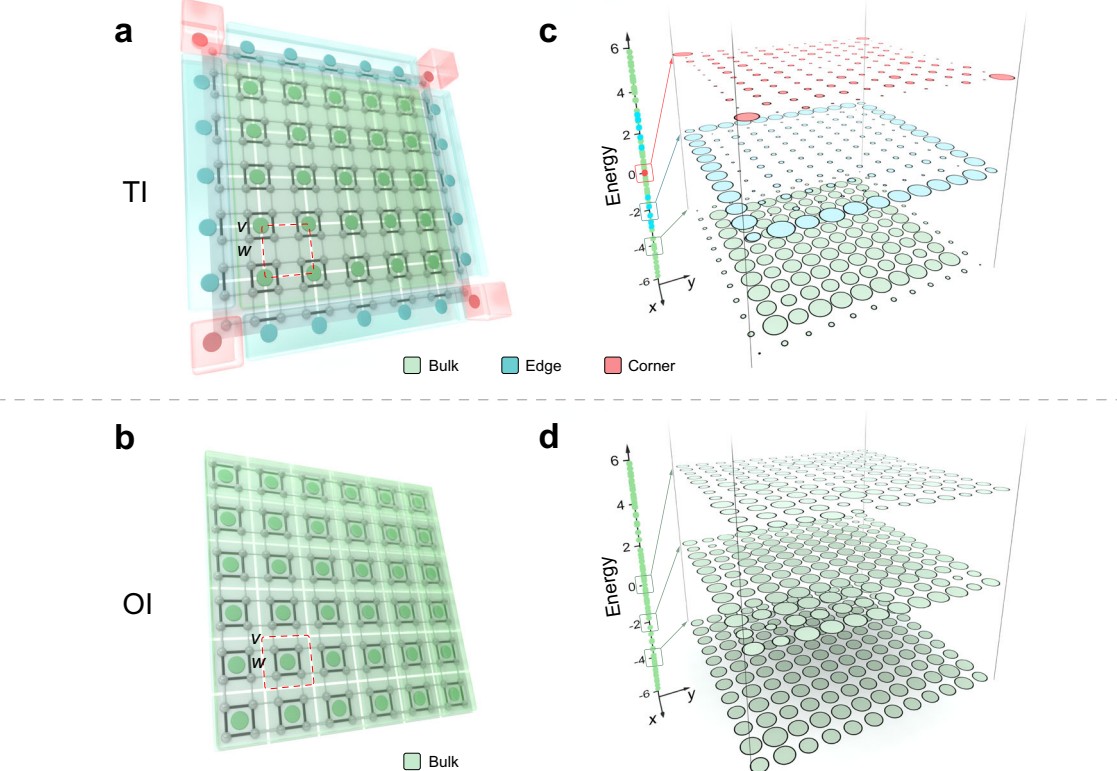

**Fig. 1 | Schematic of the bulk-LDOS correspondence. a** A topological insulator (TI) with Wannier centers (WCs) located at the corner of unit cells. The gray dots, gray lines, and black lines represent the lattice sites, intra-cell couplings and inter-cell couplings, respectively. The green, blue, and red solid circles represent WCs which are located at the bulk area (gray color area), edge area (blue color area) and corner areas (red color area), respectively. **b** A trivial atomic insulator (or ordinary insulator, OI) with WCs located at centers of unit cells. The elements have the same

meaning in (**a**). **c** and **d** The averaged local density of states (LDOS) distributions over a small energy range at different positions in the energy domain for a TI as shown in **a** and OI as shown in **b**, respectively. The green, blue, and red dots on the axis represent the eigenstates that have a higher field strength on the bulk, edge, and corner lattice sites, respectively. The sizes of green, blue, and red circles represent the magnitude of running averaged LDOS over the eigenstates inside the green, blue, and red boxes at each lattice site, respectively.

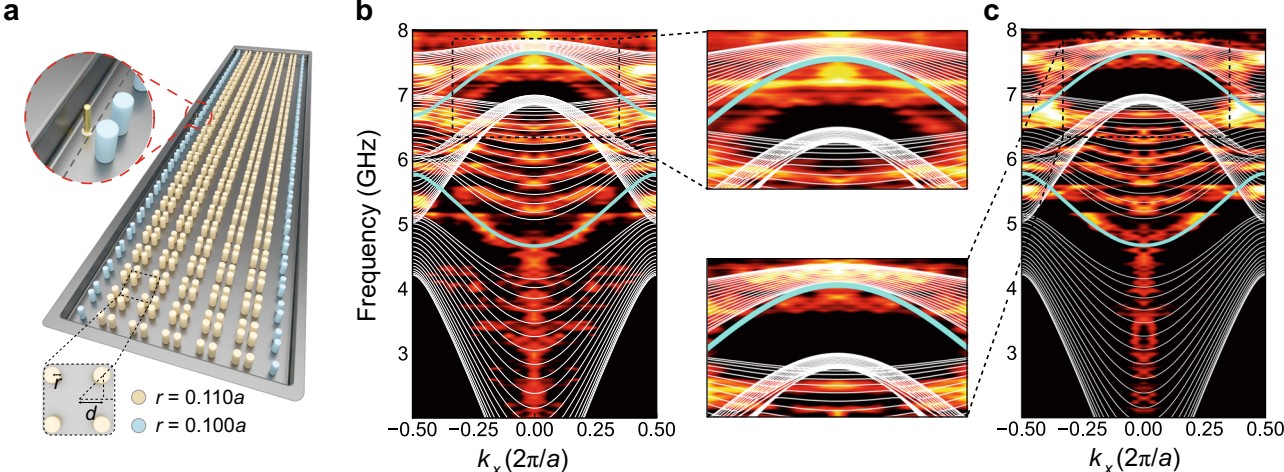

**Fig. 2 | The projected band structures (PBS) of TIs with directional bandgaps.** **a** Schematic of a finite-size photonic TI with boundary rods having a smaller radius than the bulk rods. The zoom-in inset presents the position of the probe to measure the projective band structure. The radius of the bulk and edge rods are 0.11*a* and 0.1*a* respectively. **b** The PBS of the unperturbed lattice. The white (blue) lines are the numerically simulated bulk and edge states. The bright and dark colors represent the experimentally measured PBS. The zoom-in inset shows the 1D edge state is an in-gap state. **c** The PBS of the perturbed lattice. The elements have the same meaning as in (**b**). The zoom-in inset shows the 1D edge state is embedded into the bulk spectrum for the second band gap.

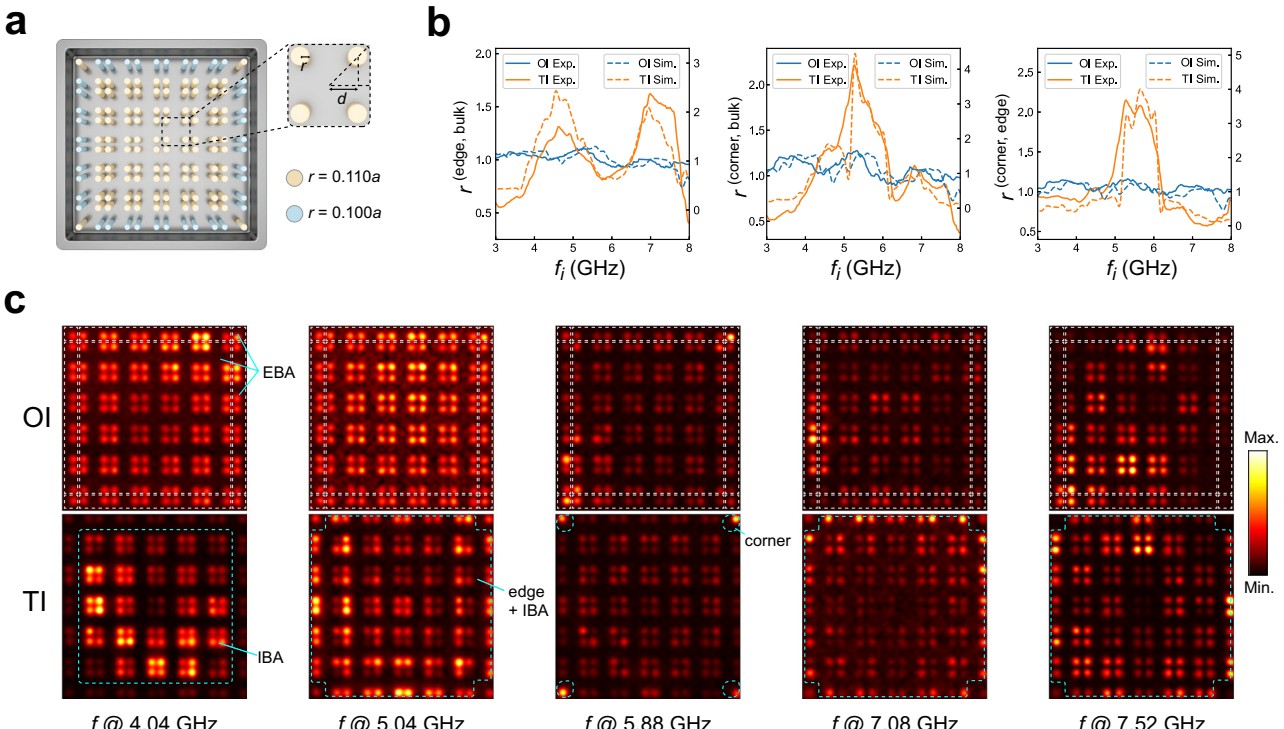

**Fig. 3 | Multidimensional partition of photonic LDOS of a TI with perturbed boundary.** **a** A finite-size photonic TI. The parameters are the same as those in Fig. 2a, b. The ratio between the running averaged edge LDOS over the running averaged bulk LDOS (left panel), the ratio between the running averaged corner LDOS over the running averaged bulk LDOS (left panel), and the ratio between the running averaged corner LDOS over the running averaged edge LDOS (left panel) are presented respectively. The solid (dashed) blue and orange lines represent the measured (simulated) ratio for the OI and TI, respectively. **c** The upper (lower) panels show the LDOS of a single frequency at 4.04, 5.04, 5.88, 7.08, and 7.52 GHz for OI and TI, respectively. For the ordinary insulator (OI), the LDOS is homogeneously distributed over the entire bulk area (EBA) at all frequencies. For the topological insulator (TI), at 4.04, 5.04, 5.88, 7.08, and 7.52 GHz, the LDOS is dominant at the interior bulk area (IBA), edge and IBA, corner, edge and IBA, edge and IBA, respectively (encircled by dashed blue lines) which clearly reveals the multidimensional partition of the LDOS.

spectrum. However, for topologically trivial lattice, all lattice sites' orbitals (represented by green area in Fig. 1b) hybridize to form bulk states and therefore the LDOS is distributed in single-dimension (here is 2-dimension for example). This multidimensional partition of LDOS in a system with open boundary conditions is an intrinsic physical property of the topologically non-trivial phase and it recovers to the conventional BBC when there is a complete bandgap with in-gap localized boundary states and FCA when there is a complete bandgap

with crystalline symmetries[33]. Moreover, it can characterize topologically non-trivial phases even in systems with directional bandgap and no in-gap boundary states or without any symmetries. The displacement of WCs in unit cells which is the bulk topological invariant has a one-to-one correspondence to the multidimensional partition of LDOS in a system with open boundary conditions, clearly revealing an intrinsic bulk-LDOS correspondence of topological phases (see detailed discussions on the tight-binding models and the difference between excited field distributions and the local density of states in Section I in SI).

Although we can directly apply the bulk-LDOS correspondence to distinguish different topological phases, in a finite-size structure with a few unit cells, it is not always clear to see the multidimensional partition of LDOS at an arbitrary energy magnitude due to the discrete resonating mode in the whole structure. To overcome this problem, we note that in a finite-size structure with open boundary conditions, different-dimensional hybridized states have different consecutive level spacings (the frequency distances between frequency-adjacent modes)[40], therefore the bulk, edge, and corner states are inhomogeneously distributed in the energy domain (see the dots in the vertical axis in Fig. 1c) for TIs. In terms of the LDOS, the running average of the LDOS for eigenmodes over a small energy range is also distributed in different dimensions in the energy domain (see Fig. 1c). This running average of the LDOS over a small energy range will reduce the finite-size fluctuations of LDOS for a single-frequency mode while preserving the character of the multidimensional partition of LDOS. However, for OIs, the running average of the LDOS over a small energy range is single-dimensional distributed instead (see Fig. 1d).

## Bulk-LDOS correspondence in topological directional bandgap materials

To experimentally observe this correspondence, we consider a 2D topological photonic crystal (PC), comprised of dielectric cylinders. PCs with engineered photonic band structures have been used to explore various topological phases including quantum Hall states[22], quantum spin Hall states[23], higher-order topological insulators[41], and with fractional quantum numbers[32]. It is noteworthy that Wannier centers (WCs) and spectral charges can also be defined within photonic bands. Prior investigations have demonstrated that by retrieving $S_{11}$ parameters and concurrently considering the Purcell effect during near-field scanning, we can directly obtain the local density of states (LDOS) for photonic states (for an in-depth discussion, refer to the "Methods" section).

In this context, we focus on a 2D photonic Su–Schrieffer–Heeger (SSH) model featuring a directional bandgap within the $s$-wave band structure (see band structure in SI). By adjusting the intercell coupling strength $t_{\text{inter}}$ and the intracell coupling strength $t_{\text{intra}}$, the 2D photonic SSH model experiences a topological phase ($t_{\text{inter}} > t_{\text{intra}}$) or a trivial phase ($t_{\text{inter}} < t_{\text{intra}}$). Moreover, the 2D photonic SSH model has been previously shown to have hierarchical topological phases with both 1D edge states and 0D corner states[41]. The higher-order topological phase in this PC can be numerically classified by the displacement of WCs of photonic modes and explained by the filling anomaly. To observe the 1D edge states, we place an excitation source at the 1D interface between the topologically nontrivial configuration and the perfect electric conductor (PEC) boundary realized by metals (see Fig. 2a). The projected band structure with in-gap 1D edge states can be seen from the Fourier transform of the excited edge states (see Fig. 2b). We then modify boundary sites by reducing the diameters of rods (see Fig. 2a). A small perturbation on the onsite energy of the boundary sites will not destroy the topological phases but only shift the frequency of edge states. Consequently, the 1D topological edge states are now embedded into the bulk spectrum as shown in Fig. 2c and there are no in-gap states in the directional bandgaps.

Now we fabricate a 2D sample with identical parameters to those in Fig. 2 to investigate the hierarchical topological phases (the detailed sample parameters are discussed in the "Methods" section) as shown in Fig. 3a. We then apply the near field scanning method by putting a metal probe near the top of the sample and extracting the $S_{11}$ parameters to obtain the LDOS of photonic states (see discussions on the measurement of photonic LDOS in the "Methods" section). We consider a structure of photonic crystals with modified boundary rods, perfect electric conduction boundary condition, and $6 \times 6$ unit cells (see Fig. 3a). The topological phases and bandgaps of this structure can be adjusted by the inter-cell couplings and intra-cell couplings which is determined by the distances between rods. Specifically, we realize two photonic OI and TI with directional bandgap and topological boundary states embedded in the bulk spectrum similar to those in Fig. 2 (see detailed discussion in SI). The detailed design of these photonic crystals is discussed in the "Methods" section. According to Fig. 2, there are no complete bandgaps from the spectra of eigenmodes and therefore one can not apply the direct measurement of spectral charges to distinguish topological phases.

Nevertheless, by applying the same approach as discussed in the preceding section, we can experimentally obtain the spectral multi-dimensional partition of photonic LDOS as shown in Fig. 3b, c for both non-trivial and trivial configurations. To provide a clear characterization of this multidimensional partition of the LDOS, we define a ratio $r$ between the averaged LDOS over a certain energy range $\Delta f$ starting from an initial frequency $f_i$ at edge (corner) sites $D^i_{\text{edge}}$ ($D^i_{\text{corner}}$) and those at the bulk sites $D^i_{\text{bulk}}$ as $r^{(n,m,i)} = \frac{D^i_n}{D^i_m}$. Here $n, m =$ bulk, edge, corner. We set $\Delta f = 1/8\Sigma$ where $\Sigma = f_t - f_b$ represents the energy range from the bottom of the band structure $f_b$ to the top of the band structure $f_t$. Here this running averaging range $\Delta f$ is chosen to ensure that the finite-size fluctuation of LDOS is reduced. We plot $r^{(\text{edge,bulk})}$, $r^{(\text{corner,bulk})}$ and $r^{(\text{corner,edge})}$ with $f_i$ starting from $f_b$ to $f_t - \Delta f$ as shown in Fig. 3b, respectively, for both TIs and OIs.

We clearly see two peaks of $r^{(\text{edge,bulk})}$ and one peak of $r^{(\text{corner,bulk})}$ and $r^{(\text{corner,edge})}$ for the TI and small fluctuations of $r^{(\text{edge,bulk})}$, $r^{(\text{corner,bulk})}$ and $r^{(\text{corner,edge})}$ for the OI. We here emphasize that unlike the tight-binding model, there exist higher-order couplings in our PCs, which inevitably break the chiral symmetry of the system. Nevertheless, the multi-dimensional partition of running averaged photonic LDOS is preserved and hence it captures general cases without any stringent requirement on the chiral symmetry of the system. From the single frequency LDOS as shown in Fig. 3c, we find that for OI (the upper panels in Fig. 3c), when we increase the frequency from the bottom to the top of the band structure, the LDOS is distributed across the 2D entire bulk area (EBA) without any partition. However, for the TI, we observe that the LDOS is distributed across the 2D interior bulk area (IBA, avoiding 1D edges and 0D corner sites) at 4.04 GHz, both 2D IBA and 1D edges (avoiding 0D corners) at 5.04 GHz, both 2D IBA and 0D corners (avoiding 1D edges) at 5.88 GHz, and both 2D IBA and 1D edges (avoiding 0D corners) at 7.08 and 7.52 GHz as shown in lower panels in Fig. 3c. This character clearly reveals the multi-dimensional partition of the LDOS for the topologically nontrivial phase.

The bulk-LDOS correspondence finds applicability in systems where no distinct detuned and separated components exist, similar to the traditional BBC and FCA for completely gapped systems[33]. Nevertheless, when such defective boundary localized states emerge by adding a large onsite energy potential to the boundary sites, we can still distinguish between topologically trivial and non-trivial cases by examining the local density of states (LDOS) in both the energy domain and real space. Specifically, there are two fundamental characteristics in the LDOS: (a) The topological boundary states stem from the hybridization of two (or even more) bands near the bandgap, and therefore, in our case, there are only two sets of edge states, and 1 set of corner states at certain frequency range while this is not the case for defect boundary localized states. For

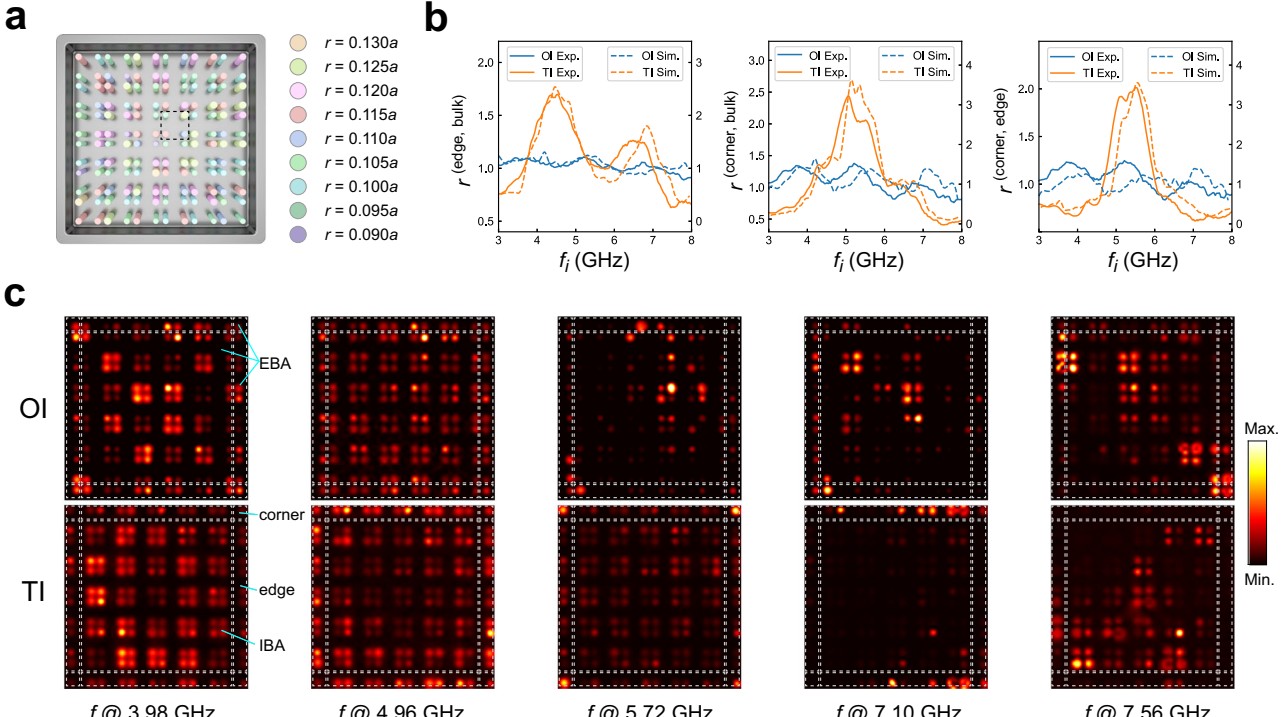

**Fig. 4 | Multidimensional partition of photonic LDOS of a TI with random disorders. a** A photonic TI with discretized random disorders on the radius of the rods. The rods with the same color have the same radius. The specific values of the radius of rods are presented. **b** The ratio between the running averaged edge LDOS over the running averaged bulk LDOS (left panel), the ratio between the running averaged corner LDOS over the running averaged bulk LDOS (left panel), and the ratio between the running averaged corner LDOS over the running averaged edge LDOS (left panel) are presented respectively. The solid (dashed) blue and orange lines represent the measured (simulated) ratio for the disordered OI and TI, respectively. **c** The upper (lower) panels show the LDOS of a single mode at $f_i = 3.98$, 4.96, 5.72, 7.10, and 7.56 GHz for disordered OI and TI, respectively.

defect states, they are introduced by the detuning process and originate from the single band. Therefore, when we add a large onsite energy potential to boundary sites in the trivial lattices, there will be more sets of boundary localized states than the topological cases. (b) In real space, the LDOS of topological boundary states are strongly localized at the boundary sites and dominant at the same sub-lattice sites. However, for the LDOS of the defect boundary states, they are extended to other lattice sites within different sublattices. A detailed discussion on the difference between topological boundary states and defective boundary states in terms of the LDOS is provided in Section IV in SI (see detailed discussions on the tight-binding models and the difference between excited field distributions and the local density of states in supplementary materials).

**Bulk-LDOS correspondence in disordered topological materials**
The Bulk-LDOS correspondence roots in the redistribution of the positions of WCs when there is a topological phase transition. Global spatial symmetries and chiral symmetry only determine the fractionalization of charges and the frequency of corner states, respectively. A small perturbation that does not close the band gaps will not destroy the relevant topological phases. Hence, our proposed bulk-LDOS correspondence also holds for topological phases with certain levels of random disorders where one cannot define the band structures and the symmetry-protected fractionalization of spectral charges. To elucidate this correspondence, we add random perturbations of on-site energy by randomly changing the diameters of the rods in the unperturbed lattice (with the same parameters as those in Fig. 2b). For simplicity in fabrication, we add the random perturbation on diameters of rods in a discrete manner. Specifically, based on the finite-size structure as discussed in Fig. 3a without modifying the boundary sites, we change the diameters of rods by a number $\lambda$ which is randomly and discretely distributed between 0 and $1/6r_0$ where $r_0 = 0.11a$ is the radius of the unperturbed rods. Under the small perturbations, there is still no complete bandgap in the spectra. Similarly, as before, we can measure the multidimensional partition of running averaged LDOS $r^{(\text{edge,bulk})}$, $r^{(\text{corner,bulk})}$ and $r^{(\text{corner,edge})}$ as shown Fig. 4b. We find there are two peaks for $r^{(\text{edge,bulk})}$ indicating the first-order topology with two non-degenerate edge states and one peak for $r^{(\text{corner,bulk})}$ and $r^{(\text{corner,edge})}$, indicating the second-order topology with corner states. We can also see the bulk-LDOS correspondence from the single frequency measurement of LDOS for the disordered structures as shown in Fig. 4c. The upper and lower panels in Fig. 4c correspond to the LDOS of wave functions for OI and TI with disorders respectively. We also see a single-dimensional partition of LDOS for OI and a multidimensional partition of LDOS for TI. We here note that due to the random disorders in the structure, the LDOS of single mode may have no $C_4$ symmetry as shown in Fig. 4c and therefore the quantized fractional charges do not exist[33]. Nevertheless, the bulk-LDOS correspondence holds on. Besides, this perturbation will also break the bound-state in the continuum (BIC) and therefore one cannot use an excited source to map out the eigenfield distributions. The excited field distributions may change significantly with respect to different positions of the source and therefore cannot be applied to diagnose the bulk topology (see detailed discussions on the tight-binding models and the difference between excited field distributions and the local density of states in Section III in SI).

## Discussion

In conclusion, we have proposed and demonstrated a rigorous Bulk-LDOS correspondence in TIs in which the topological phases are characterized by the displacement of WCs. We find that in

topologically trivial structures, the LDOS of bulk modes extend all the way to the edges and corners, while in topologically nontrivial structures the LDOS of bulk modes actually avoids the edges and corners. Our approach based on the measurement of the multidimensional partition of LDOS can identify both the first-order topological insulating phases and higher-order topological phases simultaneously. Moreover, compared to previous BBC and FCA approaches, the proposed bulk-LDOS correspondence extends its applicability to more general scenarios, including TIs featuring directional bandgaps without in-gap states and even those affected by disorders. While our experimental work is grounded in the domain of photonic crystals, the implications of our findings are broad and can be extrapolated to diverse systems, encompassing electronic materials[42], acoustic crystals[43], mechanics[44], and plasmonics[45]. Besides, we expect further exploration of this correspondence in Wannier-nonrepresentable TIs such as fragile topological phases[46] and topological phases characterized by a non-zero Chern number[26] and even gapless topological systems such as the Weyl and Dirac semimetals[47]. In practical applications, considering that the LDOS has previously been linked to the spontaneous emission rate of atoms and molecules embedded within photonic crystals[48], our discoveries offer valuable insights for future investigations into topological light emissions and light–matter interactions within directional topological bandgap materials.

Note added: During the peer-review process of our work, we became aware of a concurrent independent study on revealing topological phases in gapless systems using K-theory[49]. In comparison, our proposed method considers a more generalized model that does not require the system to possess chiral symmetry. In addition, our approach has the potential to be applied to higher dimensions.

## Methods

### Measurement of projected band structure

The projected energy band was measured utilizing a 2-ports transmission spectrum $S_{21}$ using a microwave network analyzer Keysight E5063A. We put the sample on a displacement stage with a step size of 2 mm and placed a metallic plate right above the sample to prevent radiative loss. The air gap between the sample and the upper plate is 2 mm. Two antennas are included in this measurement: one is fixed through the sample to work as an excitation source and the other is embedded in the upper plate with a distance to the PCs of 1 mm for detection. Therefore, both the amplitude and phase corresponding to the eigenmode at different positions of the sample can be measured with the moving displacement stage. During each moving step, a pause of 0.5 s is set for stationary measurement.

### Measurement of LDOS

The LDOS of the PCs can be derived from the one-port reflection spectrum $S_{11}$, which is accomplished by inserting an antenna into the upper metal plate with a coaxial cable of 50Ω connected to the analyzer. Between the sample and the upper metallic plate, an airgap of 2 mm is introduced. We averagely divide each unit cell to 10 × 10 parts and detect the central region of each part in a range of 2−9 GHz with a resolution of about 5 MHz (see detailed discussions on the tight-binding models and the difference between excited field distributions and the local density of states in Section II in SI).

We first calculate the extinction rate from the reflection as $E(f) = 1 − \|S_{11}(f)\|^2$. Considering the Purcell effect, we divide the measured extinction rate by the frequency squared, $D(f) = \frac{E(f)}{f^2}$, which is proportional to the density of states with each measured spacial point. To obtain the LDOS for one dielectric pillar $D(\mathbf{r}, f)$, we sum the $D(\mathbf{r}, f)$ in one-quarter of the unit cell up and normalize it as

$$D(\mathbf{r}, f) = \sum_i D_i(\mathbf{r}, f)\sigma_i \tag{1}$$

with $\int_{all} D(\mathbf{r}, f) = 1$ Here, the integral is taken over all bands. $D(\mathbf{r}, f)$ and $\sigma_i$ are the density of states and the area of the $i$th region with $i$ varying from 1 to 25, respectively (see detailed discussions on the tight-binding models and the difference between excited field distributions and the local density of states in Fig. S4 in SI).

To demonstrate the inhomogeneous distribution of LDOS, we average $D(\mathbf{r}, f)$ of each pillar belonging to corner, edge, and bulk, respectively. Then, the obtained $D_{corner}(f)$, $D_{edge}(f)$ and $D_{bulk}(f)$ are further integrated over the range of $\Delta f$ starting from a initial frequency $f_i$.

$$D_m^i = \int_{f_i}^{f_i + \Delta f} D_m(f) df \tag{2}$$

Here $m$ = corner, edge, bulk. $f_i$ goes from the bottom frequency of the band structure $f_b$ to the top of the band structure $f_t$. We here choose $\Delta f = 1/8(f_t−f_b)$ for the reason that this value is large enough to reduce the fluctuations of averaged LDOS due to the finite-size effect and small enough to ensure the summed set of eigenmodes are firstly all bulk states and then contain the edge states as $f_i$ increases so that we can observe the spectral inhomogeneity more clearly. For each $f_i$, we calculate the ratio as

$$r^{n,m,i} = \frac{D_m^i}{D_n^i} \tag{3}$$

Here $m, n$ = corner, edge, bulk. Specifically, we calculate $r^{(edge,bulk)}$, $r^{(corner,bulk)}$, and $r^{(corner,edge)}$ as shown in Figs. 3 and 4.

### Simulation

Numerical simulations are performed via the commercial finite-element simulation software (COMSOL MUTIPHYSICS). We build 3D photonic crystal models in all of the simulations for better correspondence with the experiments. Energy bands with infinite structures are calculated by setting the boundaries perpendicular to the propagation direction of the edge states as periodic boundaries and other boundaries as perfect electric conductor (PEC). For the calculation of the LDOS with a finite structure, we set PEC boundary conditions in all directions.

## Data availability

All data needed to evaluate the conclusions in the paper are present in the manuscript and Supplementary Information. The data are also available upon request from the corresponding author. Source data are provided with this paper.

## Code availability

The codes are available upon request from the corresponding author.

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

## Acknowledgements

We acknowledge useful discussions with Oubo You. This work was financially supported by the National Key R&D Program of China (Grants No. 2022YFA1404300, and 2021YFA1401103), the New Cornerstone Science Foundation, Hong Kong Research Grant Council (AoE/P-701/20, 17309021), National Natural Science Foundation of China (Grants No. 12174189 and 11834007), Stable Support Program for Higher Education Institutions of Shenzhen (No.20220817185604001) and the startup funding of the Chinese University of Hong Kong, Shenzhen (UDF01002563).

## Author contributions

B.X., P.Z., M.H.L., and S.Z. conceived the idea. B.X., Z.L., and S.M. performed the theoretical analysis. Z.L., R.H., S.J., J.H., and Y.J. did the numerical analysis. R.H. and S.J. did the experiment. P.Z., M.H.L., Z.L.W., Y.C., and S.Z. guided the project. All authors contributed to the discussion and writing of the manuscript.

## Competing interests

The authors declare no competing interests.
