## [Peer Review File · Nature Communications]

REVIEWERS' COMMENTS

Reviewer #1 (Remarks to the Author):

The authors have satisfactorily addressed my comments in the previous round. They also provided additional data to support their claims. In this regard, the readability of the manuscript is enhanced. However, before the manuscript can be accepted, some other revisions are necessary.

1. During the revision of the manuscript, another work on a similar concept has been recently published: Nature Communications 14, 3071 (2023). While my impression is that the manuscript under review considers a more generalized model with the feasibility to extend to even higher dimensions, I would like to request the authors to add detailed discussions to clearly state the differences and similarities compared to this published work. Especially, in terms of fundamentals, is there any stringent requirement on the chiral symmetry of the system?

2. In Figure 1, labels (a-d) are missing.

3. English has to be polished, such as a(n) eigenstate, in both main text and SI.

Reviewer #3 (Remarks to the Author):

I believe I now understand what the authors claim to be the novelty of this paper: the recognition that in trivial structures the bulk modes extend all the way to the edges and corners, while in nontrivial structures the bulk modes actually avoid the edges and corners. This is in fact somewhat interesting. I believe the paper could have been written in a way to make this simple point more clear, and figure R3 goes a long way to explaining what was not easily understandable from the paper.

We thank the reviewers for their helpful questions and providing insightful suggestions. Below we provide a one-to-one response to all the questions and comments. We mark the revisions in the manuscript in blue color.

Reviewer #1 (Remarks to the Author):

The authors have satisfactorily addressed my comments in the previous round. They also provided additional data to support their claims. In this regard, the readability of the manuscript is enhanced. However, before the manuscript can be accepted, some other revisions are necessary.

Reply: We thank the reviewer for his/her support of our work. We have revised our manuscript according to the reviewer's comments and questions.

1. During the revision of the manuscript, another work on a similar concept has been recently published: Nature Communications 14, 3071 (2023). While my impression is that the manuscript under review considers a more generalized model with the feasibility to extend to even higher dimensions, I would like to request the authors to add detailed discussions to clearly state the differences and similarities compared to this published work. Especially, in terms of fundamentals, is there any stringent requirement on the chiral symmetry of the system?

Reply: We thank the reviewer for this important suggestion. We have added detailed discussions to clearly state the differences and similarities compared to this published work as follows,

“During the peer-review process of our work, we became aware of a concurrent independent study on revealing topological phases in gapless systems using K-theory. In comparison, our proposed method considers a more generalized model that does not require the system to possess chiral symmetry. In addition, our approach has the potential to be applied to higher dimensions.”

2. In Figure 1, labels (a-d) are missing.

Reply: We thank the reviewer for pointing out this mistake. We have added the labels in Figure 1.

3. English has to be polished, such as a(n) eigenstate, in both main text and SI.

Reply: We thank the reviewer for this important suggestion. We have polished the English of our manuscript in both the main text and SI.

Reviewer #3 (Remarks to the Author):

I believe I now understand what the authors claim to be the novelty of this paper: the recognition that in trivial structures the bulk modes extend all the way to the edges and corners, while in nontrivial structures the bulk modes actually avoid the edges and corners. This is in fact somewhat interesting. I believe the paper could have been written in a way to make this simple point more clear, and figure R3 goes a long way to explaining what was not easily understandable from the paper.

Reply: We thank the reviewer for appreciating the novelty of our work. For clarity, at the beginning of the Discussion Section, we have added the sentence “In trivial structures, the LDOS of bulk modes extend all the way to the edges and corners, while in nontrivial structures the LDOS of bulk modes actually avoids the edges and corners.”